# Roles of High Osmolarity Glycerol and Cell Wall Integrity Pathways in Cadmium Toxicity in *Saccharomyces cerevisiae*

**DOI:** 10.3390/ijms22126169

**Published:** 2021-06-08

**Authors:** Yunying Zhao, Shiyun Li, Jing Wang, Yingli Liu, Yu Deng

**Affiliations:** 1National Engineering Laboratory for Cereal Fermentation Technology (NELCF), School of Biotechnology, Jiangnan University, 1800 Lihu Road, Wuxi 214122, China; yunyingzhao@jiangnan.edu.cn; 2Jiangsu Provincial Research Center for Bioactive Product Processing Technology, Jiangnan University, 1800 Lihu Road, Wuxi 214122, China; 7180201005@stu.jiangnan.edu.cn; 3China-Canada Joint Laboratory of Food Nutrition and Health (Beijing), Beijing Technology and Business University, Beijing 100048, China; wangjing@th.btbu.edu.cn (J.W.); liuyingli@th.btbu.edu.cn (Y.L.)

**Keywords:** *Saccharomyces cerevisiae*, cadmium toxicity, unfolded protein response (UPR), ER stress, reactive oxygen species (ROS), cell death, Hog1, Slt2

## Abstract

Cadmium is a carcinogen that can induce ER stress, DNA damage, oxidative stress and cell death. The yeast mitogen-activated protein kinase (MAPK) signalling pathways paly crucial roles in response to various stresses. Here, we demonstrate that the unfolded protein response (UPR) pathway, the high osmolarity glycerol (HOG) pathway and the cell wall integrity (CWI) pathway are all essential for yeast cells to defend against the cadmium-induced toxicity, including the elevated ROS and cell death levels induced by cadmium. We show that the UPR pathway is required for the cadmium-induced phosphorylation of HOG_MAPK Hog1 but not for CWI_MAPK Slt2, while Slt2 but not Hog1 is required for the activation of the UPR pathway through the transcription factors of Swi6 and Rlm1. Moreover, deletion of *HAC1* and *IRE1* could promote the nuclear accumulation of Hog1, and increase the cytosolic and bud neck localisation of Slt2, indicating crucial roles of Hog1 and Slt2 in regulating the cellular process in the absence of UPR pathway. Altogether, our findings highlight the significance of these two MAPK pathways of HOG and CWI and their interrelationship with the UPR pathway in responding to cadmium-induced toxicity in budding yeast.

## 1. Introduction

Cadmium (Cd) is a toxic heavy metal that can induce oxidative stress, cell death (apoptosis), DNA damage and even cancer in humans [1,2]. The main mechanism of the generation of reactive oxygen species (ROS) induced by Cd is to deplete cells’ anti-oxidant defences or to displace the redox-active metals from some essential enzymes [3,4,5]. Increased intracellular ROS levels can cause endoplasmic reticulum stress (ER stress), cell death (apoptosis) and DNA damage [6,7]. In budding yeast, the cadmium-induced cell death needs the caspase Yca1 and is dependent on glucose [8]. In addition, Cd-induced cell death is also mediated by the calcium signalling pathway [9]. For instance, Cd lead to a rapid elevation of the cytosolic calcium and the perturbation of ER calcium homeostasis can cause ER stress and, thus, induce cell death [10,11].

The main function of ER is to maintain a suitable environment for protein modification, protein folding, calcium homeostasis as well as ER-associated protein degradation (ERAD) and protein folding [12,13]. In yeast, the unfolded protein response (UPR) is trigged by the transmembrane sensor Ire1p when the increasing unfolded or misfolded proteins were accumulated in ER in response to ER stress [14]. The RNase activity of Ire1 was activated after its own oligomerization and autophosphorylation in response to ER stress. Activated Ire1 splices the intron from the mRNA of the transcription factor Hac1p, which was then efficiently translated and to activate the downstream target genes involved in protein folding, protein trafficking, cell wall architecture, ER-associated protein degradation (ERAD) and lipid biosynthesis [15,16]. In human cells, accumulated evidence has suggested that UPR was associated with occurrence of tumors [17,18]. ER stress is also accompanied by increased ROS generation, which then leads to cell apoptosis (death) through non-canonical protein kinase RNA—like endoplasmic reticulum kinase (PERK)-dependent translational control [19]. In addition, ER also serves as a major target of cadmium toxicity. In *Saccharomyces cerevisiae*, cadmium causes ER stress and activates UPR mainly through inducing the splicing of HAC1 mRNA and promoting Ire1 protein to cluster in distinct foci [20,21]. Similarly, mutants for *HAC1* and *IRE1* were both sensitive to cadmium, indicating the UPR is also essential for cadmium tolerance [22].

From a functional screen using homozygous diploid deletion mutant library, the mitogen-activated protein kinase signalling pathways of the high osmolarity glycerol (HOG) pathway and the cell wall integrity (CWI) pathway were first identified to be involved in the tunicamycin-induced UPR [23]. In addition, the HOG pathway and the CWI pathway were also essential for cadmium tolerance [22,24]. Meanwhile, phosphorylation activation of Hog1 and Slt2 could be induced by cadmium [25,26]. Hog1 plays crucial roles in regulating cellular processes in response to osmotic stress and cell wall stress [27,28,29]. Hog1 is activated by ER stress in a UPR dependent manner, and this process is mediated by Ssk1 branch [30]. They also found that the phosphorylated Hog1 could help the ER regain homeostasis through promoting HAC1 mRNA splicing in the late stage of ER stress. Persistent ER stress induces autophagy in the late phase and this process is accompanied with the kinetics of Hog1 phosphorylation, which regulate the stability of a critical autophagy protein Atg8 [30,31,32].

The yeast cell wall is essential for cells to protect against osmotic shock and mechanical stress, to establish and maintain cell shape, and to provide a scaffold for the cell-surface proteins [33]. The cell wall integrity (CWI) signalling pathway control these processes in response to various environmental challenges, such as heat stress, hypo-osmotic shock, oxidative stress, DNA-damaging agents and cell wall stress [33]. ER stress drugs, such as tunicamycin and dithiothreitol, could activate the CWI signalling pathway in a UPR independent manner [23,34,35]. It has been reported that tunicamycin-induced activation of Slt2 is mainly triggered by the Wsc1 sensor and the transcription factor Slm1, while the cell wall perturbing agents Calcofluor white (CFW) activated UPR via the putative cell wall stress receptors Mid2 and the transcription factor Swi6 [36,37]. Genetic phenotype analysis indicated that mutants for *IRE1* and *SLT2* were all sensitive to CFW and tunicamycin [23,36]. The MAPK Slt2 was also involved in the ER stress surveillance (ERSU) pathway that could delay ER inheritance and cytokinesis by altering the septin complex in response to ER stress, although this process is independent of the UPR pathway [35].

Although the roles of Hog1 and Slt2 in regulating UPR in response to ER stress and cadmium, the specific mechanism of these two MAPKs in the cadmium toxicity remains unclear. In this study, we showed that the three pathways of UPR, HOG and CWI were all required for cell survival in response to cadmium toxicity. We observed that the phosphorylation activation of Hog1 required the UPR pathway, while the activation of the UPR pathway needed Slt2. In addition, the results of subcellular localisation analysis demonstrated that Ire1 and Hac1 could regulate the localisation of both Hog1 and Slt2. These findings suggest that the UPR, HOG and CWI pathways are closely related and interact with each other in helping a cell to reduce cadmium toxicity. Our data sheds light on the specific mechanism of cadmium-induced toxicity in budding yeast.

## 2. Results

### 2.1. UPR Pathway Is Required for Cadmium-Induced Activation of Hog1

To further analyse the role of the HOG pathway in cadmium-induced toxicity, we first tested the growth ability of the mutants involved in the HOG pathway in response to cadmium stress. Since cadmium can cause ER stress in yeast cells [20,21], we also chose the ER stress drug, tunicamycin (TM), which triggers ER stress by inhibiting N-linked glycosylation [23]. As shown in Figure 1A, the mutants of two components of the HOG pathway, Hog1 and Pbs2, were both sensitive to Cd and TM, which was consistent with the results reported previously [22,33]. It was reported previously that both the Sln1 and Sho1 branches of the HOG pathway are needed for phosphorylation of Hog1 in response to Cd stress, while Hog1 is phosphorylated only through the Sln1 branch following DTT or TM treatment [26,30]. Consistent with these observations, we found that *hog1Δ*, *pbs2Δ* and *ssk1Δ* (Sln1 branch) mutants were both sensitive to Cd and TM, while *ste20Δ* and *ste50Δ* mutants in Sho1 branch were only sensitive to Cd (Figure 1A).

Next, we tested the unfolded protein response in mutants of the HOG pathway by measuring the *β*-galactosidase activity expressed by UPRE-*lacZ* reporter. We showed that neither the deletion of *HOG1* nor *PBS2* influenced the expression of UPRE-*lacZ* in response to Cd or TM compared with that of WT cells (Figure 1B), consistent with a previous observation that the HOG pathway controls TM-induced ER stress independent of the UPR pathway [38]. In addition, we also investigated whether the splicing level of HAC1 mRNA was affected by the HOG pathway. In hog1Δ mutants, HAC1 mRNA could be spliced normally as that of WT cells in response to Cd or TM stress (Figure 1C). In contrast, we showed that Hog1 phosphorylation levels were significantly increased in WT cells following Cd or TM treatment, but reduced in the absence of Ire1 and Hac1 (Figure 1D), consistent with the results of DTT and TM treatments [30]. Taken together, these results indicate that the components of the HOG pathway are essential for Cd tolerance and the Ire1 and Hac1-mediated UPR pathway contributes to the phosphorylation activation of the MAPK Hog1 in response to Cd, although the HOG pathway is not required for the activation of UPR in response to cadmium.

### 2.2. Ire1 and Hac1 Regulate the Nuclear Localisation of Hog1

Since the nuclear localisation of Hog1 is regulated by its phosphorylation level [39,40], and since Ire1 and Hac1 regulate the phosphorylated activation of Hog1 from the above results, we tried to evaluate whether subcellular localisation of Hog1 is regulated by Ire1 and Hac1 in response to TM or Cd. To achieve this, we tagged Hog1 with GFP at its C-terminus in the WT BY4741, *ire1Δ* and *hac1Δ* mutant strains. As expected, Hog1-GPF fusion protein displayed a significant nuclear relocation in the presence of 0.4 M NaCl in WT cells within 5 min (Figure 2), which was consistent with the previous study [41]. Upon ER stress, Hog1-GFP also displayed a nuclear location, as previously reported after 1 h of TM treatment [30]. However, upon Cd exposure, Hog1-GFP displayed similar cytoplasmic localisation to that observed in untreated cells after 1 h of Cd treatment, which was in contrast to the situation under osmotic stress conditions (Figure 2). Surprisingly, the levels of Hog1-GFP fusion protein accumulated in the nuclei of *hac1Δ* and *ire1Δ* mutants were much higher than in the WT cells, although there were no significant differences with or without TM or Cd treatment (Figure 2). It is worth noting that Hog1-GFP was transported to the cytosol after a 30 min treatment with 0.4 M NaCl in WT cells, while it could still accumulate in the nuclei of *ire1Δ* and *hac1Δ* mutants. The nuclear accumulation of Hog1 is based on its phosphorylation level, and Hog1 is phosphorylated upon ER stress and this is dependent on the UPR pathway [30]. However, disruption of the UPR pathway could still promote nuclear accumulation. These results indicate that much more phosphorylated Hog1 must enter the nucleus, where it functions in an unknown way, when the Ire1-mediated UPR pathway is inhibited.

### 2.3. The MAPK Slt2 Is Required for Both Cadmium- and Tm-Induced UPR Pathways

Among the five upstream receptors of Wsc1, Wsc2, Wsc3, Mid2 and Mtl1, we found that *wsc1Δ* mutant showed a significant growth defect in response to TM, while *mid2Δ* mutant showed a Cd-sensitive phenotype (Figure 3A). These results are consistent with those of previous studies [35], and also indicate that there may be different roles for the CWI pathway in Cd- and TM-induced ER stress. Since cell wall stress activates Slt2 via both Wsc1 and Mid2 sensors [42], both TM- and Cd-induced stresses are distinct from cell wall stress. Three transcription factors, Rlm1, Swi4 and Swi6, are known to be regulated by Slt2 [43,44]. Rlm1 is one of the known targets of Slt2, and Slt2 can be positively regulated by Rlm1 [45]. As shown in Figure 3A, we found that only the *rlm1* mutant displayed a growth defect on TM plates (Figure 3A), while *swi4* and *swi6* mutants were only sensitive to Cd stress.

Next, we examined the activity of UPRE-*lacZ* in the mutants of CWI pathway in cadmium stress. The deletion of SLT2 and its upstream component of BCK1 resulted in impaired expression of UPRE-*lacZ* during TM and Cd stress (Figure 3B). Interestingly, as shown in Figure 3B, the UPRE-*lacZ* activity was reduced in both *swi6Δ* and *rlm1Δ* mutants in response to Cd and TM stress. However, an additional observation indicated that *HAC1* mRNA splicing was not affected by deletion of *SLT2* in response to Cd (Figure 3C), in agreement with the response to ER stress reported previously [23]. These results indicate that Slt2 can regulate UPR, and this is dependent on Swi6 and Rlm1 transcription factors but independent of Ire1 or Hac1 during Cd- and TM-induced ER stress.

### 2.4. Phosphorylation and Expression of Slt2 Is Independent of the Ire1-Mediated UPR Pathway

Previous work showed that the phosphorylation level of Slt2 is increased by Cd through the cell surface sensor Mid2 [25], while phosphorylation of Slt2 is regulated by the Wsc1 senor following TM treatment [35]. To investigate whether UPR pathway regulated the phosphorylation of Slt2, we analysed the phosphorylation of Slt2 in hac1Δ and ire1Δ mutants. We showed that the phosphorylation level of Slt2 was not influenced in the absence of Ire1 or Hac1 in response to Cd, consistent with the results of TM treatment (Figure 4A) [35]. These results indicate that phosphorylation of Slt2 following both TM and Cd treatment was independent of the UPR pathway.

Since the expression of *SLT2* can be induced by ER stress [23], to investigate whether the expression of *SLT2* was regulated by Hac1 and Ire1 in response to Cd, we next monitored the expression of *SLT2*-*lacZ* in the WT BY4741, *hac1Δ*, *ire1Δ*, *swi4Δ*, *swi6Δ* and *rlm1Δ* strains. We found that SLT2-*lacZ* could be induced by Cd and TM stress in WT cells and the expression of Slt2-*lacZ* displayed the same response in hac1Δ and ire1Δ cells as WT cells in response to Cd and TM stress (Figure 4B) [12]. It should be noted that the induction of Slt2-*lacZ* was inhibited in both rlm1*Δ and swi6Δ* mutants upon Cd and TM stress (Figure 4B), indicating that the transcription factor Swi6 also regulates *SLT2* expression in addition to Rlm1. Collectively, these results suggest that the expression of *SLT2* is significantly induced by Cd and TM, and this is dependent on the transcription factor Swi6 and Rlm1, but independent of the UPR signalling pathway.

### 2.5. Defection of UPR Pathway Promotes Slt2 to Localise in Cytosol and Bud Neck

Since the phosphorylation of Slt2 can be activated by ER stress, and since Slt2 is required for cell growth and the activation of UPR in response to ER stress, we wondered whether the localisation of Slt2 is also regulated by ER stress. Therefore, we measured the localisation of Slt2 using GFP-tagged Slt2-GFP fusion protein in WT BY4741, *ire1Δ* and *hac1Δ* strains after treatment with Cd and TM. As shown in Figure 5, the nuclear localisation of Slt2 was not affected in wide type BY4741 cells in response to TM or Cd. Surprisingly, we found that in the *hac1Δ* and *ire1Δ* mutants, Slt2-GFP could be visualised in the nucleus, the cytosol, and especially the bud neck, and the distribution was similar in non-treated and stressed cells. Upon ER stress, the MAP kinase Slt2 plays crucial roles in regulating the localisation and morphology of the septin ring via a mechanism independent of the UPR pathway mediated by the ER transmembrane receptor riboendonuclease Ire1 [35]. Since Slt2 can regulate septin structures, cER inheritance, and cytokinesis upon ER stress, which is independent of the Ire1-mediated UPR [35], the deletion of Ire1 or Hac1 might result in a greater need for Slt2 function to regulate these processes, which has not been observed in WT cells. From this point of view, we reasoned that when Ire1 or Hac1 is absent, the MAP kinase Slt2 must be transported to the bud neck to perform some of these other unknown functions.

### 2.6. UPR, HOG and CWI Pathways Are All Required for Cadmium- and Tm-Induced ROS and Cell Death

Cadmium-induced ROS plays a significant role in cell death [8]. Cd can induce cell death in human T cells, mouse thymocytes, kidney cell lines, smooth muscle cells (A7r5), mouse liver cells and others [46]. In addition, the two MAPKs of Hog1 and Slt2 are also required for yeast mitophagy [47]. Since cadmium can generate intracellular ROS, to investigate whether UPR, HOG or CWI pathways was involved in the cadmium-induced ROS and cell death, we analysed ROS and cell death, we measured the intracellular ROS levels and cell death in the related mutants of UPR, HOG and CWI pathways. Herein, we showed that the levels of ROS and cell death were both induced in WT, *hac1Δ*, *ire1Δ*, *hog1Δ*, *pbs2Δ*, *slt2Δ* and *bck1Δ* cells in response to Cd or TM, and TM-induced ROS and cell death levels were higher than Cd-induced levels (Figure 6). In particular, intracellular ROS and cell death levels for the above mutant cells were all higher than those of WT cells when treated with Cd or TM, and even under the untreated control conditions. These findings suggest that Ire1-mediated ER stress, and the two MAPK pathways of HOG and CWI are all required for the induction of ROS and cell death in response to cadmium and TM. In addition, ROS and cell death levels in *slt2* and *bck1* mutants were significantly higher than in *hog1* and *pbs2* mutants in response to cadmium, Tm and even in the untreated cells (Figure 6), indicating that CWI pathway might play more important roles in regulating ROS levels and cell death.

## 3. Discussion

The two MAPK pathways play crucial roles in resistance to various stress. For instance, HOG pathway participates in hyper-osmotic stress response and the ER stress response [30,48], while the CWI pathway is involved in cell wall integrity stress, ER stress, hypo-osmotic stress response, and inheritance regulation [35,36,37]. In this report, we have shown that the HOG and CWI pathways play significant roles in the cadmium-induced toxicity. We propose that the activation of the MAP kinase Hog1 in response to Cd and TM stress is partly dependent on the Ire1-mediated UPR response, while the Cd- and TM-mediated activation of the MAP kinase Slt2 is completely independent of the UPR response. We also showed that the splicing of mHAC1 is neither dependent on Hog1 nor Slt2. Moreover, Slt2 kinase, but not Hog1 kinase, is required for the induction of the UPR in response to cadmium, which was similar to the induction of TM-induced UPR signalling as described previously [22]. In addition, in response to ER stress, the MAPK Hog1 is phosphorylated and helps the ER regain homeostasis through promoting HAC1 mRNA splicing [30]. Herein, we confirmed that Slt2 was also involved in UPR recovery along with Hog1 kinase based on measuring *HAC1* mRNA splicing levels in WT, *hog1Δ* and *slt2Δ* strains (Appendix A).

In humans and other mammals, the kidney and liver are considered as two major target organs for cadmium-induced toxicity [49]. The oxidative stress induced by cadmium can damage mitochondria and, thus, disturb the respiratory activity and cause a series of cell injuries, which then induce apoptosis or necrosis and lead to renal dysfunction or even cell death [50]. On the other hand, cadmium also triggers renal injury through causing ER stress which can induce apoptosis in response to cadmium stress [51]. Since the phosphorylation of two mammalian MAPK family members, including the extracellular signal-regulated kinase (ERK)-1/2 and p38 MAPK, can be induced by cadmium, and since the cadmium-induced phosphorylation of yeast p38 MAPK Hog1 is partially dependent on the UPR pathway, we speculate that ERK and p38 MAPK might also be correlated with the UPR pathway in the cadmium-induced toxicity in the kidney.

It has been reported previously that the Wsc1 sensor is required for cell growth, activation of the UPR, and phosphorylation of Slt2 in response to TM stress [35], while the Mid2 sensor is required for activation of the CFW-induced UPR response and cell survival in response to CFW [36]. We showed that the transcription factor Slm1 is required for Tm-induced UPR process, while Swi6 is required for the CWI-induced UPR process. In the present study, we showed that Cd-induced UPR response is also mediated by Mid2, and both Swi6 and Slm1 are required for UPRE-*lacZ* expression. These results indicate that although Cd shares some of the same characteristics with TM- and CFW-induced UPR, it also has some unique characteristics.

The MAP kinases Hog1 and Slt2 usually accumulate in the nucleus, where they activate some target transcription factors in response to stimuli, but they may have cytoplasmic substrates and function in the cytoplasm. It has been reported that Slt2 and Hog1 remain in the cytoplasm, although they are both phosphorylated during mitophagy [47]. Upon osmotic stress, Hog1 is activated by Pbs2 through two pathways, one mediated by the Sho1 branch and the other mediated by the Ssk1 branch [52]. Activated Hog1 can be imported into the nucleus to regulate some transporters that activate some target genes to help cells adapt to stress [39,40]. When the stress is eliminated, dephosphorylated Hog1 returns back to the cytoplasm [53]. Interestingly, we showed that dephosphorylated or sparsely phosphorylated Hog1 could stay in the nucleus in both *ire1* and *hac1* mutants, even when there was no stress. This suggests that Ire1 and Hac1 play important roles in importing Hog1 from the nucleus to the cytoplasm, and the dephosphorylated or low phosphorylated Hog1 might perform some unknown roles in the nucleus. Slt2 localises mainly in the nucleus, however, a polarised localisation of Slt2 at bud tips might also play significant roles. For instance, it has been reported that blocking the bud tip localisation of Slt2 via Slt2 mutation of Slt2-LA can suppress the ER inheritance defect of *ptc1Δ* mutant cells [54]. ER stress impacts the cell cycle by altering spectin stabilisation, blocking cytokinesis, and delaying ER inheritance, all of which are independent of the Ire1-mediated UPR pathway, but dependent on the MAP kinase Slt2 [35]. Herein, we demonstrated that Slt2-GFP fusion protein showed both nucleus and bud tip localisation in *ire1* and *hac1* mutants. From this perspective, Ire1 and Hac1 might also be involved in maintaining ER inheritance, cytokinesis, and septin structure along with the MAP kinase Slt2, although they might be not essential for these processes.

Our studies are important to understand the cadmium-induced toxicity. In budding yeast, cadmium can activate the UPR pathway, as well as the protein kinases of Hog1 and Stl2, which are all essential for preventing cells from cell death caused by cadmium. The UPR pathway is required for the phosphorylated activation of Hog1 but not Slt2 in response to cadmium stress. Even so, Slt2 could regulate UPRE-lacZ expression through two transcription factors of Swi6 and Rlm1. Therefore, we speculate that, in response to cadmium toxicity, the UPR pathway performs its role of inhibiting cell death partially through activating Hog1, while Slt2 performs its role partially through activating the UPR pathway. *IRE1* or *HAC1* deletion severely inhibited the activation of the UPR pathway, and also reduced the phosphorylation level of Hog1. In this case, in order to reduce the damage to cells caused by the lack of UPR pathway, more activated Hog1 must be kept in the nucleus and more activated Slt2 might be needed to transport to the cytosol or the bud neck, to perform some crucial functions.

## 4. Materials and Methods

### 4.1. Yeast Strains and Growth Media 

All *S. cerevisiae* strains used in this study are listed in Table 1. For nonselective conditions, yeast cells were grown in YPD medium, which was prepared by mixing 2% (*w*/*v*) glucose, 2% (*w*/*v*) tryptone and 1% (*w*/*v*) yeast extract. For the selection and maintenance of plasmids, an SC (synthetic complete) medium (0.17% (*w*/*v*) yeast nitrogen base, 2% (*w*/*v*) glucose, 0.5% ammonia sulfate) lacking appropriate nutrients was used. Solid media were prepared by adding 2% (*w*/*v*) agar to YPD or SC media when necessary. All yeast strains were cultured at 30 °C. Cadmium chloride and DTT were purchased from Sangon Biotech (Shanghai, China). TM and Dihydroethidium were purchased from Sigma (Beijing, China). Annexin V was purchased from Solarbio (Beijing, China).

### 4.2. Phenotype Assay 

To investigate the grown differences of mutant cells in response to cadmium and TM, the serial dilution assay method was performed as described previously [56]. Before analysis, mutant cells were first cultured overnight in liquid YPD medium, and then cells were dotted onto YPD, YPD plus 100 µM CdCl_2_, or YPD plus 2 µg/mL TM after the 10-fold serially dilution with ddH_2_O. All cells were cultured at 30 °C for 2 to 3 days.

### 4.3. DNA Manipulations

To construct the Hog1-HA and Slt2-HA fusion proteins, we amplified the 3HA- HIS3MX6 cassettes from the pFA6a-3HA-HIS3MX6 plasmid [57] with primer pairs, Hog1-CF/Hog1-CR or Slt2-CF/Stl2-CR, which were then transformed into BY4741 (wild type, WT), *hac1Δ*, and *ire1Δ* strains and integrated into the 3′ end of *HOG1* or *SLT2*, respectively. The correct transformants expressing Hog1-HA or Slt2-HA fusion protein were confirmed by PCR with primers Hog1-check-F/Hog1-check-R and Slt2-check-F/Slt2-check-R, respectively. To construct the *SLT2-lac*Z reporter, we first amplified the promoter of *SLT2* with primers SLT2-LF/SLT2-LR and replaced it with the *PMR1* promoter in pRS316-*PMR1-lac*Z [56], yielding pRS316-*SLT2-lac*Z.

To investigate the localisation of Hog1 and Slt2, Hog1-GFP and Slt2-GFP fusion proteins were constructed in the WT BY4741, *ire1Δ*, and *hac1Δ* mutant strains, respectively. To achieve this, the GFP-HIS3MX6 cassette was first amplified with primers HOG1-CGFP-F/HOG1-CGFP-R or SLT2-CGFP-F/SLT2-CGFP-R from plasmid pFA6a-GFP-HIS3MX6 [57], respectively, and integrated into the WT BY4741, *ire1Δ*, and *hac1Δ* mutant strains. Correct integration was confirmed by PCR with primers Hog1-check-F/Hog1-check-R and Slt2-check-F/Slt2-check-R, respectively. The primers were all listed in Appendix A.

### 4.4. Microscopy Assay 

C-terminally GFP-tagged Hog1-GFP and Slt2-GFP were used to visualise Hog1 or Slt2 localisation, respectively. Log phase cells expressing Hog1-GFP fusion protein were first treated with or without 0.4 M NaCl for 5 or 30 min, 50 µM CdCl_2_ and 1 µg/mL TM for 1 h, respectively. To analyse the subcellular localisation of Slt2-GFP fusion protein, log phase cells expressing Slt2-GFP were first treated with 50 µM CdCl_2_ or 1 µg/mL TM for 2 h, respectively. All cells were visualised using a Nikon Eclipse 80i epifluorescence microscope.

### 4.5. Intracellular ROS and Cell Death Assay

Yeast cells were first cultured overnight in YPD medium, then inoculated into fresh SC medium to an initial OD_600_ of ~0.1. When cells had grown to middle log phase, they were split into three aliquots, and cultured in the absence or presence of 50 µM CdCl_2_ or 1 µg/mL TM for an additional 4 h. Next, ~5 × 10^6^ cells were harvested by centrifugation and used to analyse the intracellular ROS levels and cell death. To measure the intracellular ROS levels, cells were first resuspended in 250 μL of phosphate-buffered saline (PBS) with 2.5 μg/mL dihydroethidium (DHE, sigma, Beijing, China) and incubated in the dark for 30 min. Annexin V (Solarbio, Beijing, China) staining analysis was used to test the cell death rates in response to Cd and performed as previously described [58]. All samples were visualised by fluorescence microscopy using a Nikon Eclipse 80i epifluorescence microscope and positive cells were counted by Image J software. At least 500 cells were counted in each sample, and the data are presented as the mean ± standard deviation (SD) from three independent samples.

### 4.6. β-Galactosidase Activity Assay

To measure the UPRE-driven *β*-galactosidase activity, pMCZ-Y plasmid [59] was transformed into the wild type BY4741 and the related mutant cells. Transformants were first cultured overnight in SD-URA medium, then inoculated into three aliquots of fresh SC medium and grown to middle log phase, after which 50 µM CdCl_2_ or 1 µg/mL TM was added to the medium and cells were cultured for an additional 2 h when necessary. Yeast cells were harvested by centrifugation to extract total proteins and *β*-galactosidase activity was measured as described previously [56]. Data are presented as mean ± SD from six independent experiments.

### 4.7. RNA Extraction and RT-PCR Analysis

For RNA preparation, the indicated yeast cells were first treated with or without 50 µM CdCl_2_ or 1 µg/mL TM for 2 h. In the UPR recovery experiment, TM-treated BY4741, *hog1Δ* and *slt2Δ* cells were first washed once with fresh SC medium and then were incubated in fresh SC medium for an additional 6 h. Samples were taken every hour. All cells were collected and total RNA was extracted by the hot phenol method [60]. First-strand cDNA was synthesised using a Primer Script RT reagent kit (CWBiotech, Beijing, China) according to the manufacturer’s instructions. Both the uninduced and induced expression levels of *HAC1* mRNA were estimated by PCR with the primer pair, HAC1-RT-F/HAC1-RT-R (Appendix A).

### 4.8. Protein Extract Preparation and Western Blotting Analysis

To investigate the phosphorylation levels of Hog1 and Slt2, yeast cells expressing Hog1-HA and Slt2-HA fusion proteins were first grown overnight in SC medium at 30 °C. Cells were then cultured in fresh SC medium to an OD_600_ of 0.8–1.0, and treated with 50 µM CdCl_2_ or 1 µg/mL TM for an additional 2 h. To prepare protein extracts, cell cultures were first cooled in ice water for 10 min and cells were harvested by centrifugation at 4 °C, then washed once with ice-cold water. Next, cells were resuspended in ice-cold lysis buffer [23] and vortexed with glass beads 10 times for 1 min each time. The lytic samples were centrifuged at 12,000 rpm for 10 min at 4 °C. Protein concentration was measured using a protein assay kit (Sangon Biotech, Shanghai, China) before samples were boiled with sample buffer (250 mM Tris-HCl pH 6.8, 10% sodium dodecyl sulphate (SDS), 0.05% Bromophenol Blue, 50% glycerol, 7.5% DTT) for 5 min.

To perform the Western blotting assay, samples were first analysed by 10% SDS-PAGE and transformed onto nitrocellulose membranes. Phosphorylation of Hog1-HA or Slt2-HA protein was immunodetected by phospho-p38 MAPK or phospho-p44/42 MAPK antibody (New England Biolabs, Beverly, MA, USA), respectively. Expression of Hog1-HA and Slt2-HA fusion proteins was detected by HA monoclonal antibody (Sangon Biotech).

### 4.9. Statistical Analysis

The SPSS Statistics version 19.0 is used to analyse the significant differences through its paired-samples *t*-test function. A value of *p* < 0.01 is considered to be significant.

## Figures and Tables

**Figure 1 ijms-22-06169-f001:**
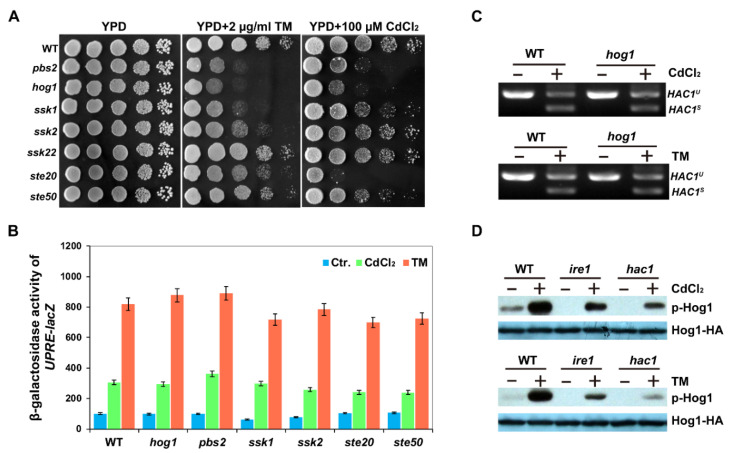
Hog1 is required for the cell growth in cadmium and Tm stresses and its activation is dependent on the UPR pathway. (**A**) Phenotypes of the mutants in the HOG pathway cadmium and Tm stress. Cells were grown overnight at 30 °C in liquid YPD and serially diluted 10-fold, then they were spotted on the indicated plates and incubated for 2–3 days at 30 °C. (**B**) *β*-galactosidase activities of UPRE-*lacZ* in wild-type BY4741 and the mutants in the HOG pathway in cadmium or Tm response. All experiments were performed in triplicate, and the error bars represent mean ± standard deviation. (**C**) *HAC1* splicing was determined by RT-PCR analysis in the wild-type and *hog1**Δ* strains treated with 50 µM CdCl_2_ or 1 µg/mL Tm for 1 h, respectively. (**D**) Phosphorylation of Hog1-HA in wild-type BY4741, *hac1Δ* and *ire1Δ* strains in response to cadmium and Tm. Cells of BY4741, *hac1Δ* and *ire1Δ* expressing Hog1-HA were grown at 30 °C in liquid YPD to log-phase and then treated with 50 µM CdCl_2_ or 1 µg/mL Tm for 1 h. Cells were then collected for protein extraction and Western blot analysis with phospho-p38 MAPK antibody or the anti-HA antibodies, respectively.

**Figure 2 ijms-22-06169-f002:**
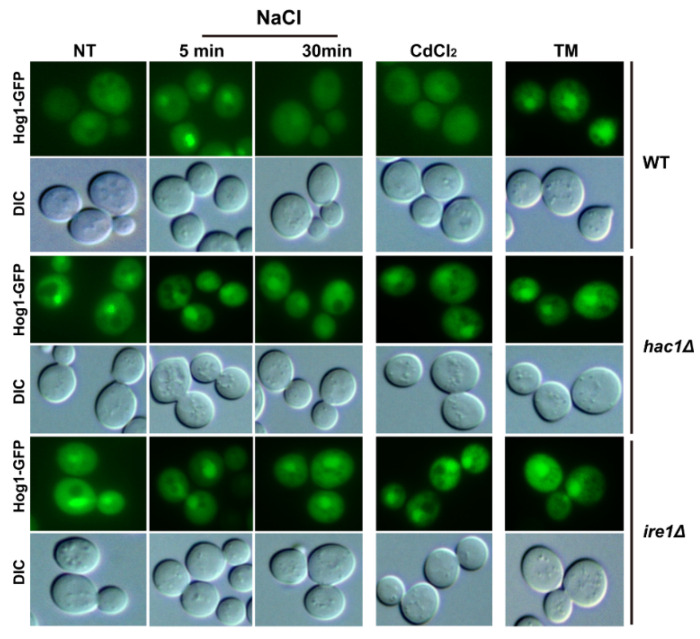
Ire1 and Hac1 regulate the nucleus localization of Hog1. Subcellular localization of Hog1-GFP in BY4741, *hac1Δ* and *ire1Δ* strains. Log-phase growing cells of BY4741, *hac1Δ* and *ire1Δ* expressing C-terminally GFP-tagged Hog1 were treated with 0.4 M NaCl, 50 µM CdCl_2_ or 1 µg/mL TM respectively. Cells were visualised under a Nikon ECLIPSE 80i fluorescent microscope.

**Figure 3 ijms-22-06169-f003:**
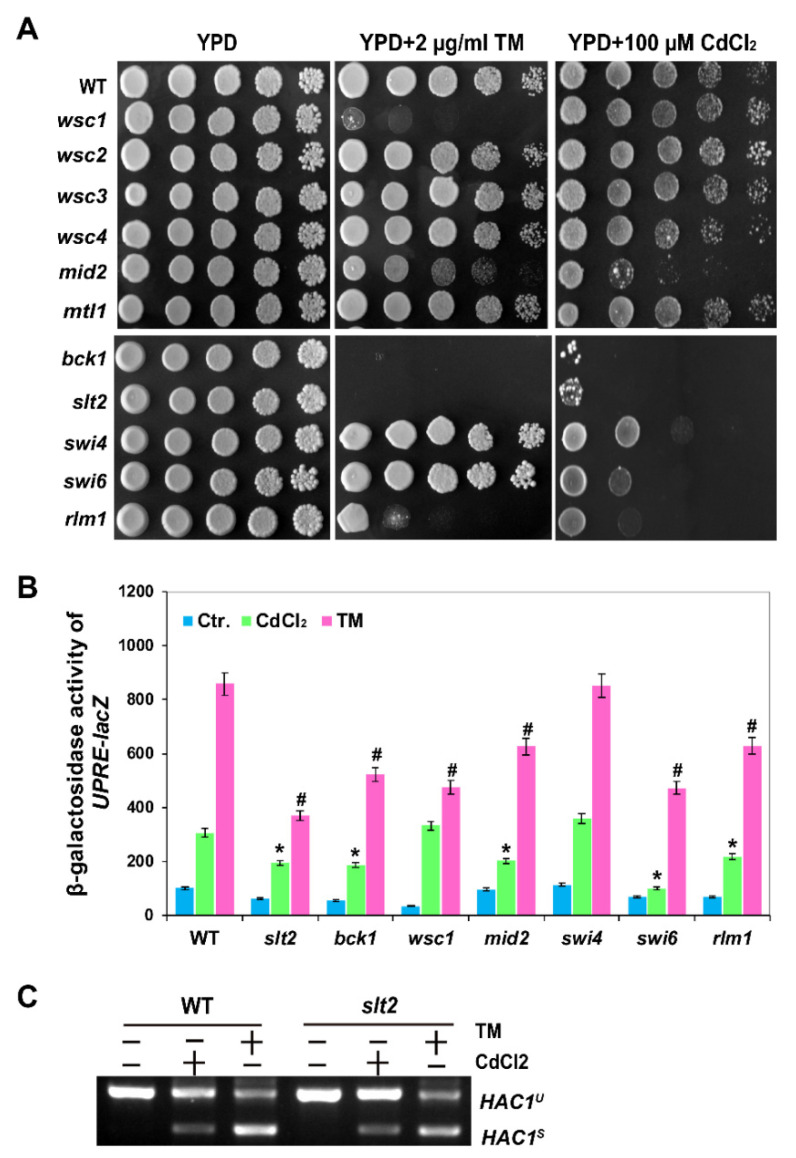
Slt2 is required for the activation of the UPR pathway. (**A**) Phenotypes of the mutants in the CMI pathway upon cadmium and TM stress. Cells were grown overnight at 30 °C in liquid YPD and serially diluted 10-fold, then they were spotted on the indicated plates and incubated for 2–3 days at 30 °C. (**B**) *β*-galactosidase activities of UPRE-*lacZ* in wild-type BY4741 and the mutants in the CWI pathway in cadmium or TM response. All of the experiments were performed in triplicate, and the error bars represent mean ± standard deviation. Results were analysed using paired-samples *t*-test function of SPSS 19.0. The significant difference of *p* < 0.01 is showed as “*” or “#” when cells were treated with cadmium or TM, respectively. (**C**) *HAC1* splicing was determined by RT-PCR analysis in the wild-type and *slt2**Δ* strains treated with 50 µM CdCl_2_ or 1 µg/mL Tm for 1 h, respectively.

**Figure 4 ijms-22-06169-f004:**
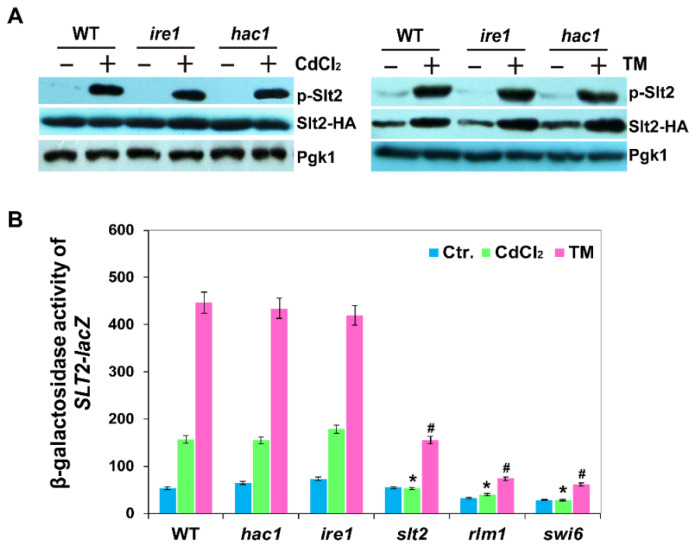
Phosphorylation and expression of Slt2 is independent of the Ire1-mediated UPR pathway. (**A**) Phosphorylation of Slt2-HA in wild-type BY4741, *hac1Δ* and *ire1Δ* strains in response to cadmium and TM. Cells of BY4741, *hac1Δ* and *ire1Δ* expressing Slt2-HA were grown at 30 °C in liquid YPD to log-phase and then treated with 50 µM CdCl_2_ or 1 µg/mL TM for 1 h. Cells were then collected for protein extraction and Western blot analysis with phospho-p44/42 MAPK antibody or the anti-HA antibodies, respectively. (**B**) *β*-galactosidase activities of SLT2-*lacZ* in response to cadmium or Tm. All of the experiments were performed in triplicate, and the error bars represent mean ± standard deviation. Results were analysed using paired-samples *t*-test function of SPSS 19.0. The significant difference of *p* < 0.01 is showed as “*” or “#” when cells were treated with cadmium or TM, respectively.

**Figure 5 ijms-22-06169-f005:**
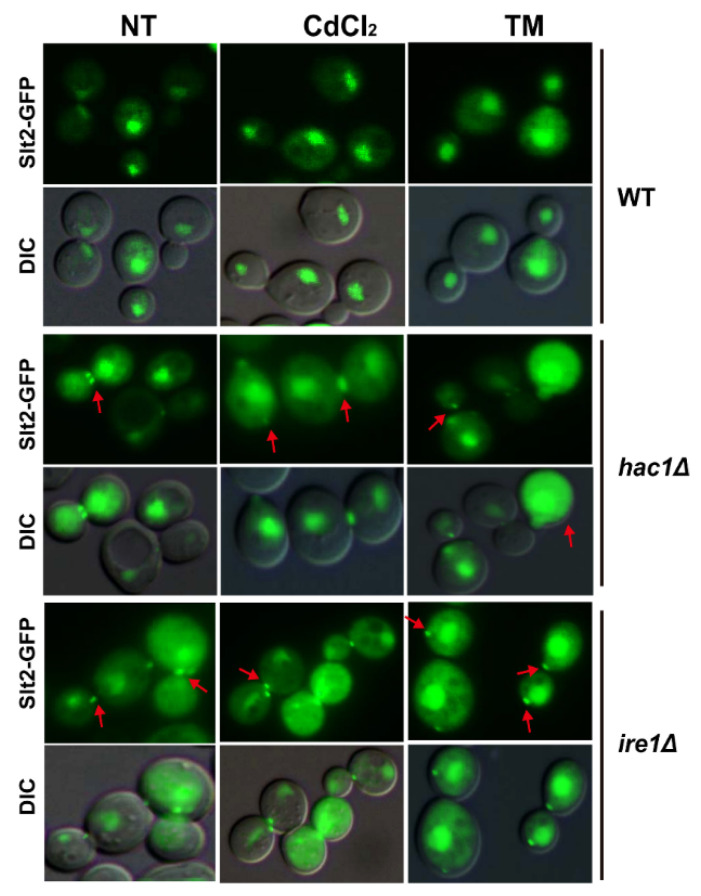
Subcellular localization of Slt2-GFP in BY4741, *hac1Δ* and *ire1Δ* strains. Log-phase growing cells of BY4741, *hac1Δ* and *ire1Δ* expressing C-terminally GFP-tagged Slt2 were treated with 50 µM CdCl_2_ or 1 µg/mL TM, respectively. Cells were visualised under a Nikon ECLIPSE 80i fluorescent microscope.

**Figure 6 ijms-22-06169-f006:**
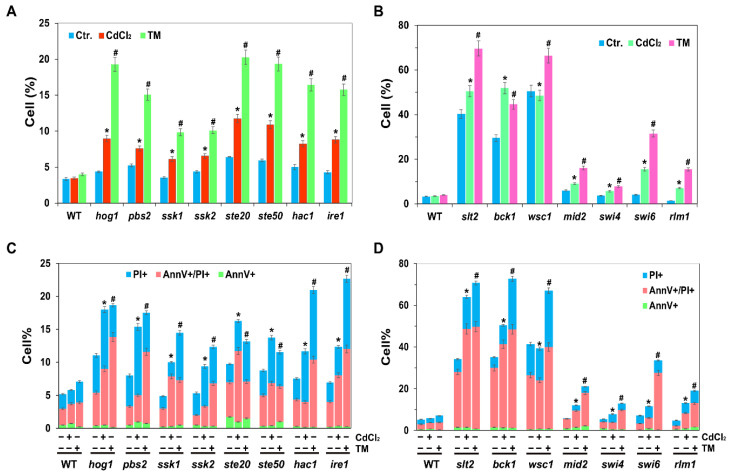
Hog1 and Slt2 are both required for the cadmium-induced and TM-induced ROS and cell death. (**A**) Intracellular ROS assay in the HOG pathway mutants. (**B**) Intracellular ROS assay in the CWI pathway mutants. (**C**) Cell death assay in the HOG pathway mutants. (**D**) Cell death assay in the CWI pathway mutants. The indicated cells were first grown to middle log phase and then were treated with 50 µM CdCl_2_ or 1 µg/mL tunicamycin for an additional 4 h. The intracellular ROS levels were tested using dihydroethidium (DHE), while cell death was measured with an Annexin V staining kit. All of the samples were visualised by fluorescence microscopy. At least 500 cells were counted in each sample, and data were presented as the mean standard deviation (SD) from three independent samples. Results were analysed using paired-samples *t*-test function of SPSS 19.0. The significant difference of *p* < 0.01 is showed as “*” or “#” when cells were treated with cadmium or Tm, respectively.

**Table 1 ijms-22-06169-t001:** *S. cerevisiae* strains used in this study.

Name	Relevant Genotype	Source/Reference
BY4741	MATa *his3Δ1 leu2Δ0 met15Δ0 ura3Δ0*	[55]
*pbs2* *Δ*	BY4741 *pbs2::kanMX4*	[55]
*hog1* *Δ*	BY4741 *hog1::kanMX4*	[55]
*ssk1* *Δ*	BY4741 *ssk1::kanMX4*	[55]
*ssk2* *Δ*	BY4741 *ssk2::kanMX4*	[55]
*ssk22* *Δ*	BY4741 *ssk22::kanMX4*	[55]
*ste20* *Δ*	BY4741 *ste20::kanMX4*	[55]
*ste50* *Δ*	BY4741 *ste50::kanMX4*	[55]
*ire1* *Δ*	BY4741 *ire1::kanMX4*	[55]
*hac1* *Δ*	BY4741 *hac1::kanMX4*	[55]
*wsc1* *Δ*	BY4741 *wsc1::kanMX4*	[55]
*wsc2* *Δ*	BY4741 *wsc2::kanMX4*	[55]
*wsc3* *Δ*	BY4741 *wsc3::kanMX4*	[55]
*wsc4* *Δ*	BY4741 *wsc4::kanMX4*	[55]
*mid2* *Δ*	BY4741 *mid2::kanMX4*	[55]
*mtl1* *Δ*	BY4741 *mtl1::kanMX4*	[55]
*bck1* *Δ*	BY4741 *bck1::kanMX4*	[55]
*slt2* *Δ*	BY4741 *slt2::kanMX4*	[55]
*swi4* *Δ*	BY4741 *swi4::kanMX4*	[55]
*swi6* *Δ*	BY4741 *swi6::kanMX4*	[55]
*rlm1* *Δ*	BY4741 *rlm1::kanMX4*	[55]
BY4741 HOG1-HA	BY4741 *HOG1-HA-HIS3MX6*	This study
*ire1**Δ* HOG1-HA	BY4741 *ire1::kanMX4 HOG1-HA-HIS3MX6*	This study
*hac1**Δ* HOG1-HA	BY4741 *hac1::kanMX4 HOG1-HA-HIS3MX6*	This study
BY4741 HOG1-GFP	BY4741 *HOG1-GFP-HIS3MX6*	This study
*ire1**Δ* HOG1-GFP	BY4741 *ire1::kanMX4 HOG1-GFP-HIS3MX6*	This study
*hac1**Δ* HOG1-GFP	BY4741 *hac1::kanMX4 HOG1-GFP-HIS3MX6*	This study
BY4741 SLT2-HA	BY4741 *SLT2-HA-HIS3MX6*	This study
*ire1**Δ* SLT2-HA	BY4741 *ire1::kanMX4 SLT2-HA-HIS3MX6*	This study
*hac1**Δ* SLT2-HA	BY4741 *hac1::kanMX4 SLT2-HA-HIS3MX6*	This study
BY4741 SLT2-GFP	BY4741 *SLT2-GFP-HIS3MX6*	This study
*ire1**Δ* SLT2-GFP	BY4741 *ire1::kanMX4 SLT2-GFP-HIS3MX6*	This study
*hac1**Δ* SLT2-GFP	BY4741 *hac1::kanMX4 SLT2-GFP-HIS3MX6*	This study

## Data Availability

The data used to support the findings of this study are included within the article and the Appendix A.

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
