# Peer review of "Roles of High Osmolarity Glycerol and Cell Wall Integrity Pathways in Cadmium Toxicity in Saccharomyces cerevisiae"

_ijms, 2021, doi:10.3390/ijms22126169_

Round 1
Reviewer 1 Report
In the present version of work, Yunyinget al did a contribution towards elevated the effects of cadmium as a carcinogen thatcan induce ER stress, DNA damage, oxidative stress and 13cell death. The quality and smooth reading was not very consist for keeping up the reader. I can say that the presented model of work exploring the toxicity pressed a lot the attention for the data variety accumulation , and diminishing the application of postanalyses for data treatment and extracting the main trends and outcomes. Yes, I can understand that this work was designed for the toxicity screening of specific mechanism of cadmium-induced toxicity in budding yeast , but still how readable and usefull compare to the exsisting resuls in literature? What is efficiency then compare to the other proven and citetated articles? The end of Introduction part very fuzzy with unclear convet! - please rephrase The data, I can say a lack of data and this makes a bad and vague expration for the reader. Talking about sich studies the statistical treament based on p-values pf other statistical signifcans was mandatory. The discusion was scars, not very convincing conclustions, the plaots are not well represanted.
For the moment I can say that the paper should be on Major revision state.
Author Response
Response to Reviewer 1 Comments
Point: In the present version of work, Yunying et al did a contribution towards elevated the effects of cadmium as a carcinogen that can induce ER stress, DNA damage, oxidative stress and cell death. The quality and smooth reading was not very consist for keeping up the reader. I can say that the presented model of work exploring the toxicity pressed a lot the attention for the data variety accumulation, and diminishing the application of post analyses for data treatment and extracting the main trends and outcomes. Yes, I can understand that this work was designed for the toxicity screening of specific mechanism of cadmium-induced toxicity in budding yeast, but still how readable and usefull compare to the existing results in literature? What is efficiency then compare to the other proven and cited articles? The end of Introduction part very fuzzy with unclear convert! - please rephrase. The data, I can say a lack of data and this makes a bad and vague expiration for the reader. Talking about such studies the statistical treatment based on p-values of other statistical significant was mandatory. The discussion was scars, not very convincing conclusions; the plots are not well represented.
Response: Thank you for all your kind suggestions and well comments. Firstly, we have reorganized each section in the “Results” section and the main trends and outcomes have been all extracted in the revised manuscript. Changes can be seen with “Track Changes” function. Secondly, in this study, we focus on demonstrating that three pathways of UPR, HOG and CWI are closely related and interact with each other in helping cell to reduce cadmium toxicity. Specifically, we found that more activated Hog1 must be kept in nucleus and more activated Slt2 might be needed to transport to the cytosol or the bud neck, to perform some crucial functions when UPR pathway was inhibited. Thirdly, the end of Introduction part have been reorganized in the revised manuscript. Changes can be seen with “Track Changes” function. Fourthly, SPSS Statistics version 19.0 software is used to analyze the significant differences through its paired-samples T-test function. A value of P<0.01 is considered to be significant. Lastly, the conclusions have been also reorganized in the last paragraph of “Discussion” section in the revised manuscript. Changes can be seen with “Track Changes” function.
Reviewer 2 Report
Authors well described one of the molecular pathways involved in cadmium toxicity. Various previous studies reported that ER stress and ROS are involved in cadmium toxicity. However, the precise pathways have been remained unknown. Authors significantly elucidated MAPK, HOG1 and SLT2 pathways are involved cadmium induced ROS and ER stress pathway. This manuscript can provide valuable knowledge for understanding cadmium toxic pathway. The experimental data enough support the hypothesis.
Cadmium is known for inducing renal toxicity, especially proximal tubule cells. Therefore, I am just wondering the pathways elucidated in this current study contribute renal toxicity. Although authors introduces the association of this current pathways in various cell lines in discussion session, if possible, it had better the involvement in the human renal function or renal dysfunction.
Author Response
Response to Reviewer 2 Comments
Point: Authors well described one of the molecular pathways involved in cadmium toxicity. Various previous studies reported that ER stress and ROS are involved in cadmium toxicity. However, the precise pathways have been remained unknown. Authors significantly elucidated MAPK, HOG1 and SLT2 pathways are involved cadmium induced ROS and ER stress pathway. This manuscript can provide valuable knowledge for understanding cadmium toxic pathway. The experimental data enough support the hypothesis.
Cadmium is known for inducing renal toxicity, especially proximal tubule cells. Therefore, I am just wondering the pathways elucidated in this current study contribute renal toxicity. Although authors introduces the association of this current pathways in various cell lines in discussion session, if possible, it had better the involvement in the human renal function or renal dysfunction.
Response: Thank you for all your kind suggestions and well comments. We have added the related discussion about the cadmium-induced toxicity in the human renal function or renal dysfunction in “Discussion” section in the revised manuscript. Changes can be seen with “Track Changes” function.
Round 2
Reviewer 1 Report
Thank you for the new updated version.
I accepted all the comments and reorganization of the paper, but I do not see any statistical data presented in the text to support the last Statistical part.
Author Response
Response to Reviewer 1 Comments
Point: Thank you for the new updated version.
I accepted all the comments and reorganization of the paper, but I do not see any statistical data presented in the text to support the last Statistical part.
Response: Thank you for all your kind suggestions and well comments. Here, SPSS Statistics version 19.0 software is used to analyze the significant differences through its paired-samples T-test function. A value of P<0.01 is considered to be significant. We have added the statistical analysis method in “Materials and Methods” (4.9 Statistical analysis-The SPSS Statistics version 19.0 is used to analyse the significant differences through its paired-samples T-test function. A value of P<0.01 is considered to be significant). In the figure legends of Figure 3, Figure 4 and Figure 6, we also added the indicated statements: “Results were analysed using paired-samples T-test function of SPSS 19.0. The significant difference of P< 0.01 is showed as “*” or “#” when cells were treated with cadmium or Tm, respectively”.

Round 3
Reviewer 1 Report
Thank you for the replay and new corrections. I think that in form as it is, the paper could be considered for publication in IJMS.